

# Acute effects of voluntary isometric contractions at maximal shortening *vs.* ballistic stretching on flexibility, strength and jump

J. Artemi Hernández-Trujillo[1,2], María Dolores González-Rivera[3], Natalia Romero-Franco[4,5] and Jorge M. González-Hernández[6]

[1] Escuela de Doctorado Area de Ciencias de la Salud, University of Alcalá, Alcalá de Henares, Madrid, Spain
[2] BioReed Lab, Tenerife, Canarias, Spain
[3] Department of Biomedical Sciences, University of Alcalá, Alcalá de Henares, Madrid, Spain
[4] Nursing and Physiotherapy Department, University of the Balearic Islands, Palma de Mallorca, Spain
[5] Health Research Institute of the Balearic Islands (IdISBa), Palma de Mallorca, Spain
[6] Faculty of Health Sciences, EVOPRED Research Group, Universidad Europea de Canarias, La Orotava, Tenerife, Spain

Corresponding author
J. Artemi Hernández-Trujillo,
artemi.hernandez@edu.uah.es

## ABSTRACT

**Background:** Isometric training is used in sport, conventional physical activity and rehabilitation. Understandably, there is a great deal of research related to its effect on performance. It is known that the length of the muscle at the moment of contraction is a determinant of strength levels. In the literature we find research on isometric training in short muscle lengths, although it has not been studied in maximally shortened positions or the acute effects that occur after its application. Ballistic stretching (BS) is also popular in sport. Their execution involves actively reaching maximally shortened muscle positions. So far, isometric training has not been compared with protocols involving ballistic stretching. Considering the above, the aim of the present study was to investigate the effects of BS and voluntary isometric contraction at maximal shortening (VICAMS) on range of motion, strength and vertical jump.

**Methods:** The study involved 60 healthy, physically active individuals (40 and 52 years old) who were randomly assigned to three groups: BS, VICAMS and a control group (CG). To assess acute effects, before and after the intervention, active range of motion (AROM), maximal voluntary isometric force (MVIF) and countermovement jump height (CMJ) were determined.

**Results:** Time main effects and time*group interactions were found for all variables ($p < 0.001$). Between-group differences were shown for the VICAMS group after the intervention, with statistically significant higher AROM values compared to the other groups. MVIF values were also higher in the VICAMS group. Intra-group differences were observed for the VICAMS and Ballistic groups, as values on all variables increased from baseline. For the CMJ, intra-group differences showed that both the VICAMS and BS groups improved values compared to baseline values.

**Conclusions:** The application of VICAMS induced acute improvements over BS in AROM, MVIF and CMJ. These results are important for coaches seeking immediate performance improvement and offer an optimal solution to the warm-up protocol.

## INTRODUCTION

Isometric training is used in sport, conventional physical activity, and rehabilitation (*Oranchuk et al., 2019*). For this reason, it is understandable that there is a great deal of research related to the effects of isometric training on strength development (*Oranchuk et al., 2019*; *Lum & Barbosa, 2019*). This type of isometric contraction, in which the muscle-tendon unit actively maintains a constant length, has been used primarily as an effective method for strength development (*Folland et al., 2005*) and as a reliable measurement of force (*Wilson & Murphy, 1996*). Retrospectively, contraction duration (*Young, McDonagh & Davies, 1985*), contraction velocity (*Maffiuletti & Martin, 2001*), tendon adaptations (*Burgess et al., 2007*) and muscle lengths have been studied (*Noorkõiv, Nosaka & Blazevich, 2014*). The muscle length at which the isometric contraction is performed is a determinant of strength levels (*Folland et al., 2005*), muscle thickness (*Alegre et al., 2014*), blood pressure and heart rate (*Ng et al., 1994*), fatigue (*Fitch & McComas, 1985*), and muscle-tendon ratio (*Kubo et al., 2006*).

It is known that in muscle contractions at short joint angles, strength gains are especially due to neural adaptations (*Noorkõiv, Nosaka & Blazevich, 2014*). For this to occur, it is essential that gamma motor neurons stretch the intrafusal fibers so that muscle spindles can send information to the central nervous system (*Kandel et al., 2013*). Otherwise, the response could be altered in these vulnerable positions and could lead to injury (*López-Chicharro & Fernández-Vaquero, 2006*). In this line of knowledge, manual tests have been used to detect neuroproprioceptive vulnerability based on the subjective assessment of strength in positions of maximum muscle shortening (*Blasco, Bernabé & Berbel, 2015*). However, isometric strength has been traditionally measured by manual dynamometry with reliability, accuracy and ease of application (*Schaubert & Bohannon, 2005*; *Bohannon, 1986b*). Two measurement strategies have been used for this purpose: *break test*, where the examiner's force exceeds that of the examinee causing an eccentric contraction (*Bohannon, 1988*); and *make test*, where the examiner offers resistance in a fixed position while the subject exerts a maximal force against the dynamometer (*Sisto & Dyson-Hudson, 2007*). It should be noted that the latter has been found to be the most reliable of the two (*Stratford & Balsor, 1994*).

BS is an example of training in which the full joint range is actively achieved and, as a consequence, the muscle is stretched to its maximum and then shortened to its maximum (*Behm et al., 2015*; *Gesel et al., 2022*). This type of stretching has been popularly included in the warm-up of athletes for many years (*Bacurau et al., 2009*; *Oliveira et al., 2017*), despite being recommended by coaches who were unaware of the effect on their performance (*Samuel et al., 2008*). In this regard, ballistic stretching has shown to increase the ability to develop strength through increased neuromuscular activity (*Hough, Ross & Howatson, 2009*), post-activation potentiation (PAP) (*Maloney, Turner & Fletcher, 2014*) and muscle temperature (*Fletcher, 2010*). Despite this, there is a historical controversy regarding the

acute effect of ballistic stretching on sports performance (*Opplert & Babault, 2018*; *Gesel et al., 2022*). Thus, we find authors who defend improvements after its application (*Maloney, Turner & Fletcher, 2014*; *Baudry & Duchateau, 2007*), authors who show no significant changes (*Unick et al., 2005*; *Alemdaroğlu, Köklü & Koz, 2017*; *Jaggers et al., 2008*) and in contrast authors who have demonstrated performance deterioration (*Nelson & Kokkonen, 2001*; *Sá et al., 2015*). Similarly, it should be noted that there is a lack of consensus in the literature when it comes to describing the execution procedures, and their eventual inclusion together with dynamic stretching in the reviews (*Behm & Chaouachi, 2011*; *Opplert & Babault, 2018*). Due to the discrepancy found in the literature, and the similar characteristics, the combination of ballistic and dynamic maximal velocity stretching is justified in further research.

In the literature we can find research on isometric training in short muscle lengths of the lower body (*Bandy & Hanten, 1993*; *Bogdanis et al., 2019*), although it has not been studied in positions of maximum shortening, nor the acute effects that occur after its application in a single session. Similarly, isometric training has not been compared with protocols that include ballistic stretching. Therefore, the main purpose of this study was to compare the acute effects of ballistic stretching (BS) and voluntary isometric contraction at maximal shortening (VICAMS) as part of a warm-up on range of motion, strength and vertical jump of physically active population. We hypothesized that effects of VICAMS will induce higher active range of motion (AROM), maximal voluntary isometric force (MVIF) and countermovement jump (CMJ) height compared to BS.

## MATERIALS AND METHODS

### Design

A randomized controlled repeated measures design was used to evaluate and compare the acute effects of VICAMS and BS on active range of motion, maximal isometric strength and vertical jump. After 2 weeks of familiarization, participants were randomly assigned into three groups: a BS group ($N = 20$), a VICAMS group ($N = 20$) and a control group (CG) ($N = 20$). Each group consisted of 10 males and 10 females. In order to assess the acute effects, before and after the intervention, AROM, MVIF and CMJ were determined. AROM and MVIF were measured for hip flexion (HF) and knee flexion (KF). The CG was re-measured 20 min after the first measurement but did not undergo any intervention.

### Participants

Sixty healthy, recreationally active participants (30 females and 30 males; age 46.4 ± 5.5 years old; height 1.70 ± 0.17 m; weight 76.0 ± 14.9 kg; body mass index (BMI) 25.5 ± 2.5 kg/m$^2$) were voluntarily enrolled in this study. All participants were instructed not to engage in physical activity the day before the test.

Inclusion criteria were: (1) to be between 35 and 55 years old; (2) to practice non-competitive physical activity at least twice a week. Exclusion criteria was: having a musculoskeletal injury or disease that prevented the correct performance of the tests. All participants received a detailed explanation of the study procedures and signed the informed consent form. The study was approved by the Research Ethics and Animal
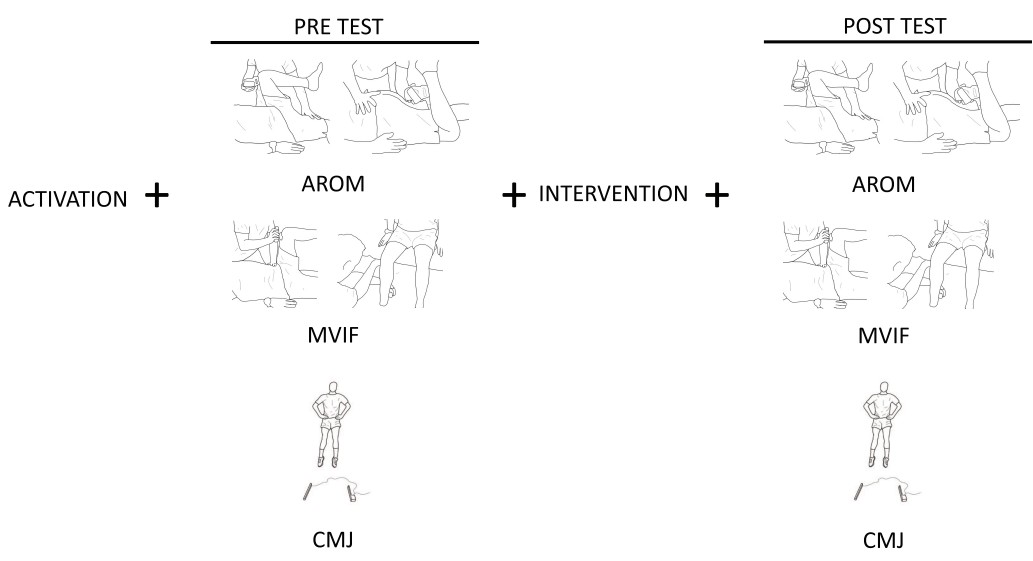

**Figure 1 Scheme of study design.** The order in which the study was carried out and the variables that were measured are shown: Active range of motion (AROM), maximal voluntar isometric force (MVIF), counter movement jump (CMJ).               

Experimentation Committee of the University of Alcalá (CEID/2022/4/083) and conducted in accordance with the principles established by the Declaration of Helsinki.

## Testing procedures

To ensure a high level of reliability, all measurements were performed by the same researcher, who has more than 5 years of experience. The temperature in the laboratory was kept constant at around 23 °C. AROM and MVIF were measured with a calibrated handheld digital goniometer-dynamometer model Micro-FET3 (Hoggan Scientific, Salt Lake City, Utah, USA). This device provides reliability and ease of measurement (*Bohannon, 2012*). CMJ was measured with a 420 × 590 mm Chronojump contact platform (Boscosystem), which allows reliable and cost-effective measurement of jump height and changes in physical performance (*Pueo, Penichet-Tomas & Jimenez-Olmedo, 2020*).

 Prior to the measurements the participants performed a standard warm-up of five raises of each knee, five abductions of each hip, five extensions of each hip, five bends of each knee and 10 heel raises at a time. All in a standing position, slow and controlled, and reaching the position of maximum muscle shortening. The same exercises were then repeated in a ballistic manner. To finish, they did five squats and three CMJ. The measurements were then performed in the following order: (1) AROM (3-min pause); (2) MVIF (3-min pause); (3) CMJ (5-min pause); (4) Intervention (5-min pause); (5) AROM (3-min pause); (6) MVIF (3-min pause); (7) CMJ. A scheme of the study design is shown in Fig. 1.

 Active range of motion (AROM): Measurements of bilateral lower extremity AROM were carried out according to the methods described by *Norkin & White (1986)*. Thus, HF was measured in the supine position and KF in the prone position. The correct starting

position was then established with the goniometer, coinciding with the anatomical position. The subject was then asked to perform the movement in a slow and controlled manner to the maximum possible joint range. At this final position, the measurement was recorded. Three measurements were taken for each joint position and recorded from the digital readout on the goniometer. The highest of the three trials was used for data analysis. After each measurement, the limb was returned to the starting position. A 10-s rest was provided between measurements to minimize the effect of fatigue, and 30 s between positions to accommodate the posture of the participant and the investigator. The researcher observed whether compensatory movements occurred when executing the movement. If the participants made compensatory movements, they were corrected immediately. In cases where the participant was unable to correct the compensatory movement, the angle was measured at the point prior to compensation.

Maximal voluntary isometric force (MVIF): It was performed using the positions and techniques described by *Andrews, Thomas & Bohannon (1996)*. Force measurements are expressed in kilograms. The same protocol was followed as in the previous measurements, obtaining three measurements and using the highest for subsequent analysis (*Brinkrnann, 1994*). This reduced the possible learning effect. To reduce the fatigue effect, subjects rested 15 s between each measurement (*Sisto & Dyson-Hudson, 2007*) and 30 s between positions (*Arnold et al., 2010*). The positions for correct measurement were explained to the participants prior to the test. In cases where the subject did not demonstrate maximal effort, did not follow instructions or the HHD recorded an error message, the test was repeated with adequate rest compensation. To measure HF, the participant lay supine and raised one leg with the hip flexed at 90° and the knee relaxed. The examiner placed the dynamometer on the anterior side of the leg at the level of the femoral condyles and ensured that the pelvis was stabilized during the measurement. For the KF, the participant sat on the stretcher with the hips and knees flexed at 90°. The dynamometer was placed on the posterior side of the leg, just proximal to the malleoli. Trunk stability was monitored during the measurement. Subsequently, participants were instructed to exert as much force as possible against the device while maintaining the position. Meanwhile, the researcher held the device as still as possible without breaking the position (make test), as this has been shown to be more reliable than the break test (*Stratford & Balsor, 1994*). The duration of the contraction was approximately 5 s (*Thorborg et al., 2010*) and the instructions given to the participant were: push, hold, relax. It has been shown that this time is sufficient to reach maximal force (*Wang, Olson & Protas, 2002*). Participants were asked not to make an explosive effort, but to increase force progressively until peak force was reached (*Maffiuletti et al., 2016*). The device pad and the examiner's forearm were held perpendicular to the limb being measured. (*Bohannon, 1986a*; *Andrews, Thomas & Bohannon, 1996*; *Wang, Olson & Protas, 2002*).

Countermovement jump (CMJ): In the warm-up, participants were instructed to familiarize themselves with proper jumping technique and a balanced landing. The assessment was carried out following the Bosco protocol (*Bosco, Luhtanen & Komi, 1983*) and each jump was recorded in centimeters. For the measurement, three jumps were performed with 1 min rest in between to avoid the effect of fatigue (*Read & Cisar, 2001*).

The highest value was selected as final value. The execution technique consisted of placing both hands at hip level, bending the knees to a self-selected depth and attempting to jump as high as possible in a single movement. As the participants were inexperienced in CMJ, a knee flexion close to 90° was suggested. Only correctly executed jumps were considered. The jump was repeated for any mismatch that could affect the measurement. Verbal encouragement was provided to help achieve maximum effort on each jump (*Heishman et al., 2018*). Similar and consistent instructions were given to each participant prior to the test, to limit possible effects on jumping (*Young, Pryor & Wilson, 1995*).

## Intervention

Ballistic stretching protocol: Each subject performed three sets of 20 s on each leg for each exercise, with a recovery of 10 s each time. The total time of the intervention was approximately 18 min. A beat rate of 40 beats per minute measured by a metronome was set (*Samuel et al., 2008*; *Oliveira et al., 2017*). The subject's foot had to touch the ground on each repetition for full range of motion, except for knee extension, which had to return to the starting position. He was instructed to perform the movement following the metronome rhythm and reaching the maximum possible muscle shortening without bouncing (*Yamaguchi & Ishii, 2005*; *Sekir et al., 2010*). Verbal instructions were given during the execution to correct the exercise in case the execution was incorrect. It was performed first with the right leg and then with the left leg.

The exercises performed correspond to the movements measured in the ROM, AROM and MVIF. The order followed was:

1) Hip flexion (HF). The subject stood with hands resting on the hips to avoid arm assistance. He was instructed to lift one leg forward with the knee flexed and return to the starting position.

2) Hip abduction (HABD). The subject stood with hands resting on a wall to help stabilize the body during the movement. He/she was instructed to separate the leg to the side with the knee extended and return to the starting position.

3) Hip adduction (HADD). The subject stood with hands resting on a wall to help stabilize the body during the movement. He was instructed to slightly delay the supporting leg and bring the free leg forward just in front. The free leg went from neutral to maximum adduction and then to the starting position.

4) Hip extension (HE). The subject stood with hands resting on a wall to help stabilize the body during the movement. The subject was instructed to move the leg backwards with the knee extended and return to the starting position.

5) Knee flexion (KF): The subject stood with hands resting on a wall to help stabilize the body during the movement. The subject was instructed to flex and extend the knee to return to the starting position.

6) Knee extension (KE): The subject lay supine with one hip and knee flexed to 90°, holding the leg with the hands above the knee to remove the force of the hip flexors. He was instructed to extend and flex the knee to the starting position.

Voluntary isometric contraction at maximal shortening protocol (VICAMS): The joint movements selected for this protocol were the same as in the BS protocol and in the previous measurements. For the performance of each exercise, the researcher guided the participant manually and verbally throughout the application. VICAMS were performed after the researcher brought the joint to the maximum possible muscle shortening. Prior to each exercise and coinciding with the breaks, the subject was shown where the main muscles responsible for performing the movement were located so that he/she could become aware of where he/she had to exert the force. In addition, the subject was palpated with two fingers at various points along the path of these muscles. The participant was asked to concentrate on the muscles that are activated to produce the movement, and to resist by trying to maintain a constant opposing force of as similar intensity as possible to the one he was receiving. From the position of maximum shortening, the researcher exerted force in the opposite direction to the movement and the participant tried to resist this force.

The total time of the intervention was approximately 18 min, almost the same as the Ballistic Stretching. Each voluntary contraction lasted 5 s. Nine repetitions of each movement were performed with 3 s rest between repetitions, 15 between each leg and 30 between exercises. The intensity of the force applied was divided into three levels: three repetitions at light intensity (20–30%); three repetitions at medium intensity (45–55%); and three repetitions at maximum intensity (100%). It was applied in that order. After each repetition, the shortening position was maintained and insisted to reach the maximum. In order to minimize stimuli in other directions, the researcher kept the forearm perpendicular to the segment where the force was applied (*Leal, Martínez & Sieso, 2012*). Verbal cues during execution were: contract, hold, relax. The participant was allowed to hold on to the stretcher to remain stable in case of unbalance. It was performed first with the right leg and then with the left leg.

The order of the exercises was like that of the Ballistic Stretches:

1) Hip flexion (HF). In the supine position, force was applied just above the anterior aspect of the knee in the direction of hip extension, keeping the knee flexed.

2) Hip abduction (HABD). In the supine position, force was applied just above the outside of the knee in the direction of hip adduction. The weight of the leg was held with the other hand to minimize the action of gravity.

3) Hip adduction (HADD). In order to achieve maximum muscle shortening, one leg was raised just above the other. In the supine position, force was applied just above the inside of the knee in the direction of hip abduction. The weight of the leg was held with the other hand to minimize the action of gravity.

4) Hip extension (HE). In prone position, force was applied just above the posterior aspect of the knee in the direction of hip flexion. The other hand was used to help keep the knee extended and to minimize the weight of the leg.

5) Knee flexion (KF). In prone position, the force was applied just above the posterior aspect of the ankle in the direction of knee extension.

6) Knee extension (KE): The subject was seated on the stretcher with the legs hanging down, hip-width apart. The force was applied just above the anterior aspect of the ankle in the direction of knee flexion.

### Statistical analysis

Descriptive data are presented as mean and standard deviation (SD). Normality of data was checked with Kolmogorov Smirnov test. The sample size was calculated using GRANMO version 7.12 (Barcelona, Spain). Accepting a 5% significance level and 80% statistical power, 14 participants were required for every group. This sample size was necessary to obtain an improvement of 7° in AROM of hip flexion. The standard deviation considered was 6.4°, based on previous studies (*López-Bedoya et al., 2013*). Baseline characteristics of participants were compared using independent samples Student's t-tests. Since there were no between-group differences at baseline, a two-way (group × time) repeated-measures analysis of variance (ANOVA) with the Bonferroni *post hoc* test was used to evaluate group by time interactions and within-group and between-group effects. Confidence interval (CI) 95% were obtained for all differences and effect sizes (ES) were obtained in case of significant differences, interpreted by using Cohen's d, considering a small (0.1), moderate (0.3), large (0.5), very large (0.7), and extremely large (0.9) effect (*Hopkins et al., 2009*). International Business Machines (IBM) SPSS statistics, version 21.0 (Chicago, IL, 224 USA) was used, and statistical significance was set at $p < 0.05$.

## RESULTS

Table 1 shows all data regarding AROM, MVIF and CMJ variables of all participants, according to the group they belonged. Main time effects and time*group interactions were found for all the variables ($p < 0.001$).

For AROM in both right and left knees, intra-group differences were observed for VICAMS and ballistic groups, by increasing values compared to baseline ($p < 0.01$). Between-groups differences (Fig. 2) were shown for VICAMS group after the intervention, with statistically significant higher AROM values in right knee compared to ballistic ($p = 0.001$), and control group ($p < 0.001$), and higher AROM values in left knee compared to ballistic ($p < 0.001$), and control group ($p < 0.001$).

Finally, in terms of MVIF, intra-group differences were observed in all participants, with higher MVIF values in the right and left knee compared to baseline values ($p < 0.05$). Between-group differences (Fig. 3) showed that, after the intervention, the VICAMS group had significantly higher MVIF values in the right knee and left knee compared to the ballistic group ($p = 0.005$ right knee and $p = 0.01$ left knee) and the control group ($p = 0.001$ right knee and $p = 0.004$ left knee). Results between groups are shown in Fig. 3.

For AROM in both right and left hips, intra-group differences were observed for VICAMS and BS groups, by increasing values compared to baseline ($p < 0.01$). Between-groups differences (Fig. 4) were shown for VICAMS group after the intervention, with statistically significant higher AROM values in right hip compared to BS ($p = 0.003$), and CG ($p < 0.001$), and higher AROM values in left hip compared to BS ($p < 0.001$), and control group ($p < 0.001$).

**Table 1 Data of active range of motion (knee and hip), maximal voluntary isometric force (knee and hip) and countermovement jump for pre and post measurements in participants belonging to all groups.**

| | | Pre | | Post | | Intra-group differences (pre vs. post) | | | Between-group differences (at post) | | | |
|---|---|---|---|---|---|---|---|---|---|---|---|---|
| | Group | Mean | SD | Mean | SD | Mean | 95%CI | ES (d) | | Mean | 95%CI | ES (d) |
| AROM Right Knee Flexion (degrees) | Ballistic | 78.15 | 4.77 | 78.55 | 4.87 | 2.30 | [1.69–2.91] | 0.08 | Ballistic vs. Vicams | 6.35 | [2.23–10.47] | 0.80 |
| | Vicams | 78.75 | 4.85 | 82.45 | 4.91 | 7.60 | [6.16–9.04] | 0.76 | Vicams vs. control | 7.00 | [2.88–11.12] | 0.91 |
| | Control | 77.90 | 4.84 | 78.05 | 4.80 | 0.65 | [0.06–1.24] | NS | Control vs. ballistic | 0.65 | [−3.47 to 4.77] | NS |
| AROM Left Knee Flexion (degrees) | Ballistic | 77.95 | 3.59 | 78.60 | 3.55 | 1.80 | [1.36–2.26] | 0.18 | Ballistic vs. Vicams | 6.10 | [2.81–9.39] | 1.34 |
| | Vicams | 78.80 | 3.00 | 83.20 | 3.30 | 6.15 | [4.65–7.66] | 1.40 | Vicams vs. control | 7.05 | [3.76–10.34] | 1.34 |
| | Control | 77.35 | 5.30 | 77.30 | 5.27 | 0.60 | [0.04-1.16] | NS | Control vs. ballistic | 0.95 | [−2.34 to 4.24] | NS |
| AROM Right Hip Flexion (degrees) | Ballistic | 121.60 | 7.01 | 124.50 | 7.74 | 2.90 | [2.19–3.61] | 0.39 | Ballistic vs. Vicams | 7.75 | [2.28–13.22] | 0.98 |
| | Vicams | 120.35 | 8.25 | 132.25 | 8.04 | 11.90 | [10.09–3.72] | 1.46 | Vicams vs. control | 10.35 | [4.88–15.82] | 1.57 |
| | Control | 121.60 | 4.72 | 121.90 | 4.77 | 0.30 | [−0.27 to 0.87] | NS | Control vs. ballistic | 2.60 | [−2.87 to 8.07] | NS |
| AROM Left Hip Flexion (degrees) | Ballistic | 119.65 | 6.70 | 121.70 | 7.00 | 2.05 | [1.51–2.59] | 0.35 | Ballistic vs. Vicams | 11.65 | [7.07–16.23] | 1.92 |
| | Vicams | 120.85 | 6.12 | 133.35 | 4.93 | 12.50 | [10.7–14.3] | 2.25 | Vicams vs. control | 12.15 | [7.57–16.73] | 2.33 |
| | Control | 120.80 | 4.96 | 121.20 | 5.47 | 0.40 | [−0.11 to 0.91] | NS | Control vs. ballistic | 0.50 | [−4.08 to 5.08] | NS |
| MVIF Right Knee Flexion (Kg) | Ballistic | 27.14 | 3.59 | 27.86 | 3.62 | 0.38 | [0.20–0.55] | 0.2 | Ballistic vs. Vicams | 3.41 | [0.84–5.98] | 1.41 |
| | Vicams | 27.92 | 3.47 | 34.84 | 6.01 | 3.76 | [3.29–4.23] | 1.41 | Vicams vs. control | 3.95 | [1.38–6.56] | 1.33 |
| | Control | 27.94 | 3.71 | 28.13 | 3.83 | 0.14 | [−0.05 to 0.33] | NS | Control vs. ballistic | 0.54 | [−2.03 to 3.11] | NS |
| MVIF Left Knee Flexion (Kg) | Ballistic | 27.26 | 5.06 | 27.83 | 5.00 | 0.16 | [−0.01 to 0.32] | NS | Ballistic vs. Vicams | 3.67 | [0.73–6.60] | 1.32 |
| | Vicams | 27.03 | 4.06 | 34.48 | 5.07 | 3.79 | [3.40–4.18] | 1.62 | Vicams vs. control | 3.98 | [1.04–6.92] | 1.24 |
| | Control | 27.50 | 5.65 | 27.77 | 5.76 | 0.09 | [−0.09 to 0.26] | NS | Control vs. ballistic | 0.32 | [−0.62 to 3.25] | NS |
| MVIF Right Hip Flexion (Kg) | Ballistic | 15.83 | 3.27 | 16.14 | 3.34 | 0.31 | [0.09–0.53] | 0.09 | Ballistic vs. Vicams | 3.37 | [0.67–6.06] | 0.92 |
| | Vicams | 15.67 | 2.77 | 19.51 | 3.98 | 3.84 | [3.21–4.46] | 1.11 | Vicams vs. control | 3.46 | [0.76–6.15] | 0.99 |
| | Control | 15.95 | 2.81 | 16.05 | 2.95 | 0.10 | [−0.08 to 0.28] | NS | Control vs. ballistic | 0.09 | [−2.60 to 2.78] | NS |
| MVIF Left Hip Flexion (Kg) | Ballistic | 16.19 | 3.37 | 16.52 | 3.41 | 0.33 | [0.13–0.53] | 0.10 | Ballistic vs. Vicams | 4.50 | [2.00–7.00] | 1.35 |
| | Vicams | 16.47 | 2.52 | 21.02 | 3.27 | 4.55 | [4.12–4.98] | 1.56 | Vicams vs. control | 5.00 | [2.50–7.49] | 1.62 |
| | Control | 15.92 | 2.87 | 16.02 | 2.90 | 0.11 | [−0.81 to 0.29] | NS | Control vs. ballistic | 0.49 | [−2.00 to 2.99] | NS |
| CMJ (m) | Ballistic | 0.17 | 0.04 | 0.18 | 0.05 | 0.00 | [0.24–0.44] | 0.22 | Ballistic vs. Vicams | 0.00 | [−3.69 to 4.24] | NS |
| | Vicams | 0.17 | 0.06 | 0.18 | 0.06 | 0.01 | [0.94–1.64] | 0.17 | Vicams vs. control | 0.02 | [−2.23 to 5.70] | NS |
| | Control | 0.16 | 0.04 | 0.16 | 0.04 | 0.00 | [−0.12 to 0.15] | NS | Control vs. ballistic | 0.01 | [−2.51 to 5.43] | NS |

**Note:**

AROM, active range of motion; CI, confidence interval; CMJ, countermovement jump; Vicams, voluntary isometric contraction at maximal shortening; ES, effect size; MVIF, maximal voluntary isometric force; SD, standard deviation.

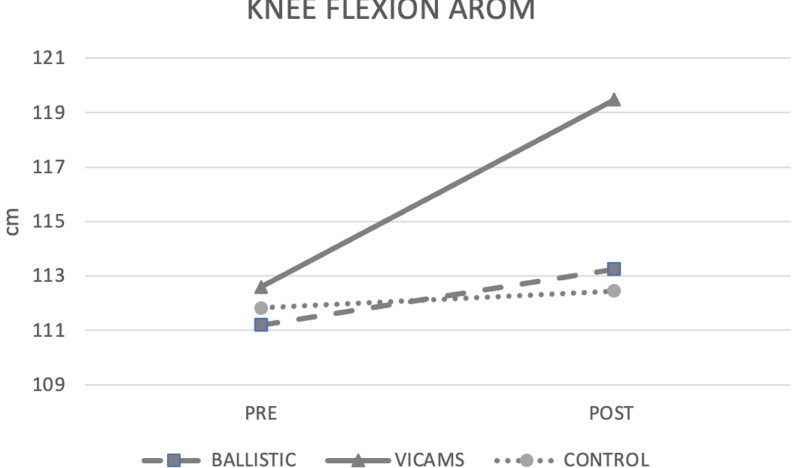

**Figure 2 Pre- and post-difference between groups.** Mean values between both knees to AROM.

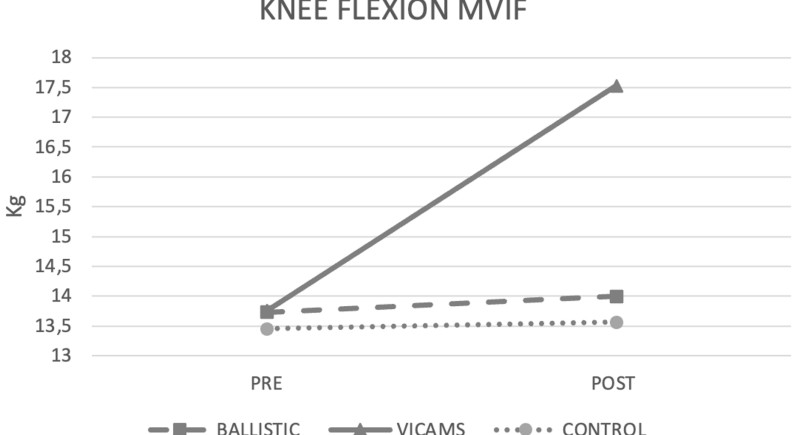

**Figure 3 Pre- and post-difference between groups.** Mean values between both knees to MVIF.

Regarding MVIF, intra-group differences were shown only for VICAMS and BS groups, with higher values compared to baseline in both right and left hips ($p < 0.01$). Between-group differences (Fig. 5) showed higher values for VICAMS group in right and left hips compared to BS ($p = 0.009$ and $p < 0.001$, respectively) and CG ($p = 0.007$ and $p < 0.001$, respectively).

For CMJ, intra-group differences showed that both VICAMS ($p < 0.001$) and BS ($p < 0.001$) groups improved values compared to baseline. No between-group differences were found for CMJ ($p > 0.05$) (Fig. 6).

## DISCUSSION

The main objective of this study was to compare the acute effects of BS and VICAMS on AROM, MVIF, and CMJ in a physically active population. Both groups showed improvements in all variables, although VICAMS were significantly greater than BS. It is

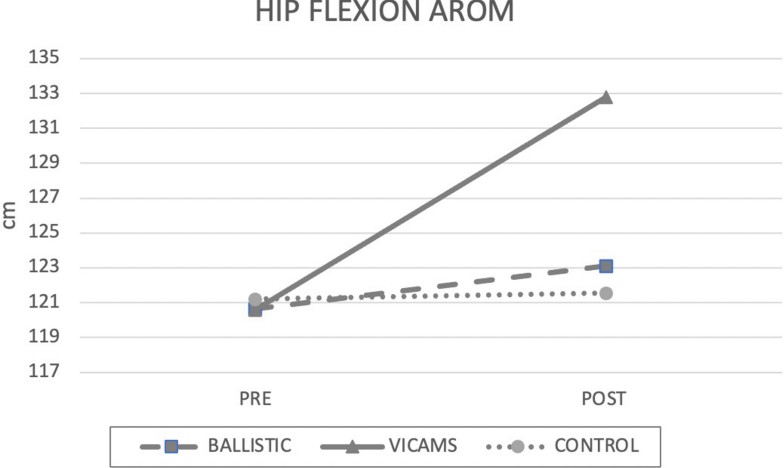

**Figure 4 Pre- and post-difference between groups.** Mean values between both hips to AROM.

important to expose the existing lack of consensus regarding the execution of BS (*Opplert & Babault, 2018*). Consequently, we can find them in the literature as rebound-type cyclic movements (*Gesel et al., 2022*; *Bandy, Irion & Briggler, 1998*), or as a form of dynamic stretching with higher velocities (*Opplert & Babault, 2018*). In either case, they are characterised by the intention to execute the active movement at maximum speed (*Desmedt & Godaux, 1977*) and in which the mass is accelerated throughout the movement (*Newton et al., 1996*). Likewise, in the literature there are studies that performed stretches that they called active, but executed in a ballistic way. *Hough, Ross & Howatson (2009)*, *Yamaguchi & Ishii (2005)*, *Fletcher (2010)*, *Sekir et al. (2010)*, *Manoel et al. (2008)*, and even with rebounds in cycles of 1:1 s (*Carvalho et al., 2012*). Moreover, given the difficulty of separating them due to their similar characteristics, they have been combined to investigate their effects on muscle performance (*Opplert & Babault, 2018*; *Behm & Chaouachi, 2011*).

The results of the present study indicated that there were improvements in AROM. This is in agreement with other research that has demonstrated immediate improvement in flexibility following BS (*De Vries, 1962*; *Konrad, Stafilidis & Tilp, 2017*; *Nelson & Kokkonen, 2001*; *Bacurau et al., 2009*; *Wiemann & Hahn, 1997*). To achieve their statements, they used procedures of different durations: 15 s (*Nelson & Kokkonen, 2001*; *Wiemann & Hahn, 1997*), 30 s (*De Vries, 1962*; *Konrad, Stafilidis & Tilp, 2017*; *Hardy & Jones, 1986*) and 1 min (*Bacurau et al., 2009*). Similarly, in our study, 20-s periods were applied, which are within the ranges used previously (*Nelson & Kokkonen, 2001*; *Konrad, Stafilidis & Tilp, 2017*; *Bacurau et al., 2009*). These improvements in flexibility can be justified by the decrease in muscle-tendon unit stiffness and passive resistive torque (*Konrad, Stafilidis & Tilp, 2017*). Furthermore, it is known that increased temperature following BS (*Bishop, 2003a*; *Yamaguchi & Ishii, 2005*; *Fletcher, 2010*) can reduce viscosity (*Buchthal, Kaiser & Knappeis, 1944*) and improve flexibility by reducing resistance to stretching (*Kubo et al., 2001*). Regarding improvements after the application of VICAMS,

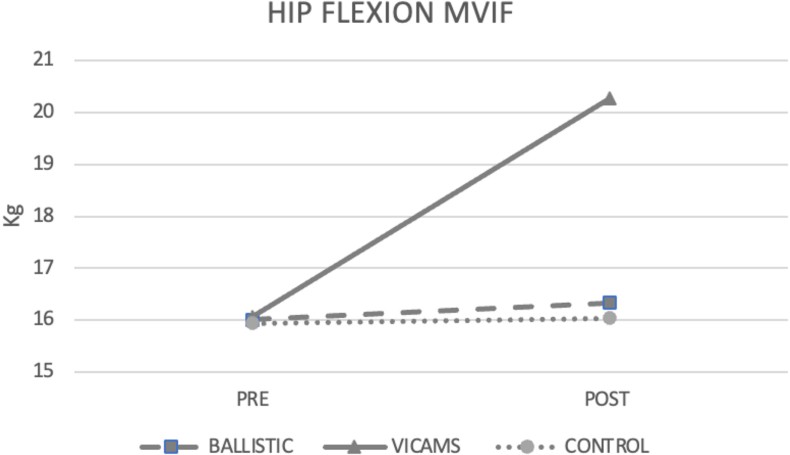

**Figure 5** **Pre- and post-difference between groups.** Mean values between both hips to MVIF.

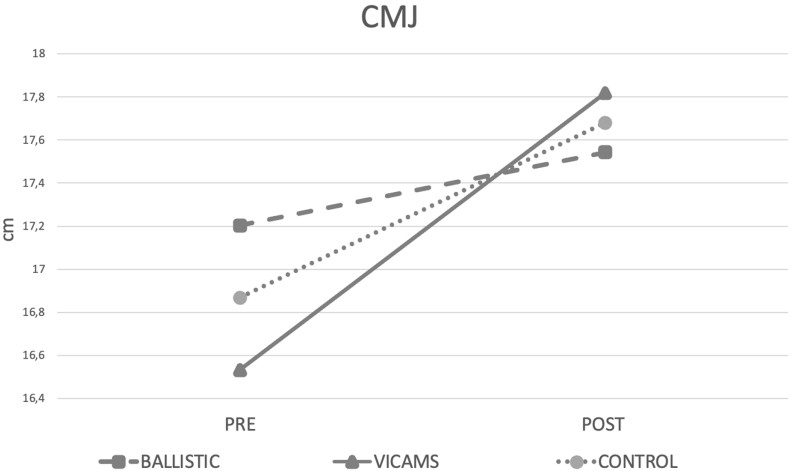

**Figure 6** **Pre- and post-difference between groups.** Mean values to CMJ.

it is known that the use of isometric contractions of the agonist muscles and the consequent relaxation of the antagonist muscles is a good way to improve active flexibility (*Hartley-O'Brien, 1980*; *Hardy, 1985*). Our results corroborate this finding, as significant improvements in AROM were obtained after its application. Among the few studies that relate the effects of isometric training on AROM, we must highlight that of *Hartley-O'Brien (1980)*, who used an active stretching system where the position was held at the end of the swing for 6 s. They also found significant improvements of more than 15° in AROM after 3 weeks of intervention. We could establish similar points to our study, in which the shortening position was held for 5 s activating the agonist muscles, although in their case, without applying an external opposing force and over a much longer period. A total of 5 years later, *Hardy (1985)* repeated the *Hartley-O'Brien (1980)* study including passive manoeuvres and obtained similar results regarding the improvement of active flexibility. Considering that the main objective of isometric training is not to generate

effects on flexibility, it is understandable that there is a lack of research in this area. If the current results of the BS and VICAMS are contrasted, the gains of the latter are greater. This may be due to the fact that the BS provokes the activation of the stretch reflex and the consequent contraction of the muscle being stretched, impairing the improvement of ROM (*Mahieu et al., 2007*). Based on the above, it can be suggested that VICAMS may be an effective method to improve AROM.

Our results indicated acute improvements in maximal isometric strength development in both the BS group (1.96%) and the VICAMS group (26.70%). Consistent with these findings, there are studies in the literature that advocate the positive effects of BS on muscle performance (*Yamaguchi & Ishii, 2005*; *Sekir et al., 2010*; *Maloney, Turner & Fletcher, 2014*; *Manoel et al., 2008*; *Hough, Ross & Howatson, 2009*; *Gelen, 2010*). In contrast, others have found neutral (*Bacurau et al., 2009*; *Samuel et al., 2008*; *Unick et al., 2005*) or even negative effects (*Nelson & Kokkonen, 2001*). Although there are studies in the literature that relate BS and strength development, no studies have been conducted to assess the acute effects of BS on through MVIF. An example that comes close is that of *Manoel et al. (2008)* who assessed the acute effects through an isokinetic knee extension test. After the application of three sets of 30 s of dynamic stretching at maximum speed, they found significant improvements in strength levels (7.6% on average). These improvements are consistent with those in our study and can be accounted for by increased neuromuscular activity (*Hough, Ross & Howatson, 2009*), PAP (*Maloney, Turner & Fletcher, 2014*), and increased temperature (*Bishop, 2003b*; *Fletcher, 2010*; *Yamaguchi & Ishii, 2005*). After the application of isometric contractions, strength gains associated with neural factors have also been verified (*Moritani & DeVries, 1979*; *Kitai & Sale, 1989*; *Sale, 1988*), as a consequence of PAP induction (*Sale, 2004*). This is consistent with our results after the application of VICAMS, and is justified by the theory that strength levels are optimized through increased contractile performance of the muscle when MVIF are included (*Bishop, 2003a*). Within this framework, although isometric training has been investigated in short muscle lengths of the lower body (*Bandy & Hanten, 1993*; *Bogdanis et al., 2019*) the acute effects have not been studied in positions of maximum muscle shortening as in the present investigation. In this line of work, *Bandy & Hanten (1993)* found strength improvements not only in the trained angle after an 8-week isometric knee extension training with 30° of flexion. This coincides with our study in which we trained at maximum shortening and strength improved at an angle other than the one trained. Following the above statements, if the current BS results are contrasted with VICAMS, the higher levels of improvement in MVIF can be justified based on two issues: on the one hand, considering that isometric contractions in a shorter length, compared to a longer one, have a lower fatigability (*Kooistra, de Ruiter & de Haan, 2008*; *Fitch & McComas, 1985*); and on the other hand that, at shorter lengths, they have greater neural activation (*Babault et al., 2003*; *Noorkõiv, Nosaka & Blazevich, 2014*). Based on the above, it can be suggested that VICAMS may be an effective method for acutely increasing muscle strength.

In this research, there were improvements in CMJ after intervention with BS (1.99%) and with VICAMS (9.13%), but there were no significant differences between the groups. These results add to the controversy regarding the acute effects of BS on specific

performance variables. On the one hand, there are authors who defend the improvement of vertical jump (*Hough, Ross & Howatson, 2009*; *Fletcher, 2013*, *2010*; *Woolstenhulme et al., 2006*), and on the other hand, authors who claim the lack of significant effects on it (*Bradley, Olsen & Portas, 2007*; *Jaggers et al., 2008*; *Unick et al., 2005*; *Samuel et al., 2008*). In line with the results of the present study, *Fletcher (2013)* studied the effect on the CMJ of two sets of 10 repetitions of dynamic stretching at maximum speed in the lower body, finding improvements in jump height after the inclusion of these after an active warm-up. Improvements were mainly attributed to PAP induction (*Fletcher, 2013*). Also, the increase in AROM could be increasing the depth of the CMJ and changing the jumping strategy to improve jump height. In contrast, the lack of acute effects of ballistic stretching on jump height has been reported. *Bradley, Olsen & Portas (2007)* reported negligible effects of BS of the quadriceps, hamstrings and plantar flexors. It is possible that the selection of these muscles for jumping improvement negatively influenced the results, as the importance of introducing gluteal muscle group activation for CMJ performance improvement has been reported (*Crow et al., 2012*). In the case of VICAMS, improvements in vertical jumping after its application are justified by increased neural activation (*Babault et al., 2003*; *Noorkõiv, Nosaka & Blazevich, 2014*) and increased PAP activation (*Vandervoort, Quinlan & McComas, 1983*; *Stuart et al., 1988*) at shorter muscle lengths, which may help in the improvement of jumping power. In the study of *Tsoukos et al. (2016)* reported CMJ improvement after six maximal isometric squats of 3 s at different knee angles, detailing greater improvements in a shorter position (140°) than neutral (90°). Based on the results, it can be concluded that VICAMS can be an effective strategy to improve the performance of explosive actions such as vertical jumping. The BS group obtained improvements, although not greater than the CG.

## Limitations

The current study has several limitations. Firstly, the profile of the participants means that the results cannot be generalized to athletes. Future research should focus on the acute and chronic effects that VICAMS can have on athletes on these same variables. Secondly, the need for a person to apply the resistances may reduce the independence of the participants. In future research, it would be interesting to use another type of resistance that can be performed without the need for another person. Thirdly, only the effect of VICAMS on the lower body has been investigated. Acute and chronic effects on the upper body should also be studied. Also, the use of jump mats instead of force plates is a limitation. In the field of vertical jump diagnostics, force plates are the gold standard (*Rogan et al., 2015*), although jumping mats are inexpensive and easier to transport. Furthermore, the study lacks ecological validity for use in athletes. Finally, another suggested line of research could be the effects of VICAMS on subjects undergoing rehabilitation.

## CONCLUSIONS

The application of VICAMS induced acute improvements above the BS in AROM, MVIF and CMJ. These results are important for coaches seeking immediate performance

improvement and give an optimal solution to warm up protocol. Given this potential, its effect should be investigated as part of training in athletes.

## ACKNOWLEDGEMENTS

The authors thank all those who participated as subjects in this study.

### Funding
The authors received no funding for this work.

### Competing Interests
J. Artemi Hernandez-Trujillo is employed by BioReed Lab.

### Author Contributions
- J. Artemi Hernández-Trujillo conceived and designed the experiments, performed the experiments, analyzed the data, prepared figures and/or tables, authored or reviewed drafts of the article, and approved the final draft.
- María Dolores González-Rivera conceived and designed the experiments, authored or reviewed drafts of the article, and approved the final draft.
- Natalia Romero-Franco analyzed the data, authored or reviewed drafts of the article, and approved the final draft.
- Jorge M. González-Hernández conceived and designed the experiments, analyzed the data, prepared figures and/or tables, authored or reviewed drafts of the article, and approved the final draft.

### Human Ethics
The following information was supplied relating to ethical approvals (*i.e.*, approving body and any reference numbers):

The study was approved by the Research Ethics and Animal Experimentation Committee of the University of Alcalá (CEID/2022/4/083).

### Data Availability
The raw measurements are available in the Supplemental File.

### Supplemental Information
Supplemental information for this article can be found online at http://dx.doi.org/10.7717/peerj.17819#supplemental-information.

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
