# Peer review of "Acute effects of voluntary isometric contractions at maximal shortening *vs.* ballistic stretching on flexibility, strength and jump"

_PeerJ, doi:10.7717/peerj.17819_

## Round 0.1 · original submission · Major Revisions

I congratulate the authors on an interesting study. Before publications, several points raised by the reviewers need to be addressed. This includes methodological concerns as well as a limitations section which needs to be added.

Reviewer 1 ·

Basic reporting

see all comments below

Experimental design

see all comments below

Validity of the findings

see all comments below

Additional comments

This article examined the performance effects of isometric contractions versus ballistic stretching. Overall, this investigation answers a current concept in the field and has a sound methodology. I believe this adds to the literature and should be accepted following minor corrections which I will detail below. The authors should be applauded for a well constructed and analyzed project.
Major comments: be consistent with terms and usage. Try, when possible, to stick with concentric, isometric, and eccentric contractions throughout. This is the most common terminology and can help with readability.
Minor:
55: need comma after activity
58: specify muscle contraction and that it’s producing force actively.
60: this is a vague statement. Do you mean it’s reliable as a testing and monitoring option?
63: this sentence “This length factor……needs to be rewritten its hard to understand what you are alluding to.
68: use concentric in place of shortened muscle contraction
84-86: re-write this sentence for clarity.
148: Is there a sampling rate associated with this mat or is it such as a just jump mat? If so please provide sampling rate
214: You state legs are bent to 90 degrees. Was this confirmed everytime? Most studies allow for self-selected depth. Can you describe why you chose to fix the depth of the CMJ
354: Change to “ Both groups showed improvement……………..
403: You can back this statement up by providing the percent change for each group.
434: I would state this is an acute increase in strength longer term was not assessed in this study.
437-459: This paragraph needs to be condensed and you need to bring in why your results were found. Why did your groups not find a significant difference? Again use % differences between groups to make your case for your results.

·

Basic reporting

The study is well written with a good level of English language used. It does have a professional structure with a good written body. However, the table has missing information, moreover as the figures only report a mean difference it is misleading, as within the text you discuss a significant difference for the experimental groups but nothing for the control group who experienced similar changes, this would have been further explained by the missing information within the table.

There was a sufficient range of references used for the text giving a clear context.

Experimental design

The article does fall into the scope of the journal, although I would say the quality of the work needs improving. The research question is well defined and relevant and would fill the gap attempted to be presented within the literature review.

There is a reasonable attempt to discuss the methods, however, there is missing information which would be useful (see specific comments).

Validity of the findings

The findings could present impact and novelty, however further exploration of the information is required to present both the strengths and limitations of the study. Even though the methods are valid, the limitation of the methods does require exploration, as it could be misleading presenting as it currently is limiting the impact.

Additional comments

General comments

It is a nice study that has merit and interesting findings, however, it has a number of flaws including consistency within the text, some of the text is confusing and needs further exploration. Moreover, highlighting the limitations within the methods used would also be crucial. You also need to take due care and attention with your referencing, as there are a number of inaccuracies within the text and the reference list missing information.

The figures and information within the table should also be improved.

Specific comments

Abstract

L24 – I am not sure if isometrics are used within conventional physical activity, can you be more specific?

L30 - Is ballistic stretching not attempting to reach a maximal muscle length?

Introduction

L60 – Change to “as a reliable measurement of force (Wilson and Murphy., 1996).”

L61 – Check reference style

L63-66 – This sentence appears to be missing something, what does this information mean for the present study.

L72 – check reference style

L84 – Delete “Within this order of ideas, we can state that”

L84 – You have already abbreviated to BS, use consistently throughout the text, this is a consistent issue.

L91 – Abbreviation of PAP needed here.

L94-97 – Why are there contrasting views?

L101-102 – What do you mean by the combination of ballistic stretching and dynamics maximal velocity shortening?

L105 – Check reference style
L109 – abbreviations, also you should not capitalise words just because they will be used as an abbreviation.

Materials and methods

L131 – Define the classification by

McKay AKA, Stellingwerff T, Smith ES, Martin DT, Mujika I, Goosey-Tolfrey VL, Sheppard J, Burke LM. Defining Training and Performance Caliber: A Participant Classification Framework. Int J Sports Physiol Perform. 2022 Feb 1;17(2):317-331. doi: 10.1123/ijspp.2021-0451. Epub 2022 Dec 29. PMID: 34965513.

L143-150 – Delete all, as this information should be in the procedures.

L160 – were the CMJs performed within the warm up maximal or submaximal?

L161-163 – Can you provide some rational to the order of testing?

L168 – reference style.

L171 – Explain what you mean by controlled movement to the position of maximum shortening? Use joint/anatomical terminology.

L173 – Why did you use the highest and not an average of the 3?

Hopkins, W.G., Measures of reliability in sports medicine and science. Sports Med, 2000. 30(1): p. 1-15. http://www.ncbi.nlm.nih.gov/entrez/query.fcgi?cmd=Retrieve&db=PubMed&dopt=Citation&list_uids=10907753

L184 – reference style

L185- reference style

L185 – If it is registered in kg, it is not force. Please amend accordingly.
L191 – How did you determine if participants did not perform a maximal effort?

L200 – Can you be sure there was no movement? Was any rest provided to the researcher performing the assessments?

L203 – Delete “peak”

L204-205 – add reference
Maffiuletti NA, Aagaard P, Blazevich AJ, Folland J, Tillin N, Duchateau J. Rate of force development: physiological and methodological considerations. Eur J Appl Physiol. 2016 Jun;116(6):1091-116. doi: 10.1007/s00421-016-3346-6. Epub 2016 Mar 3. PMID: 26941023; PMCID: PMC4875063.
L214 – Did participants always flex to 90 degrees? How was this judged?

L214 – Change “far” for “high”

L215 – Were any instructions provided for the flight? i.e. maintain full knee and hip extension, etc.Or anything that could alter flight time.

L219 – Reference style

L224 – Was the duration sufficient based on the literature?

L239-252 – Why include full lower body? Why not only knee flexion and hip flexion?

L257 – Abbreviation.

L259 – Abbreviation.

L268 – Was it a true maximal contraction if they were instructed not to overcome the force?

L274 – why was it applied in that order?

L299 – Was any a-priori sample size estimation performed?

L299 – Why did you not perform any reliability to determine if changes are meaningful?

L305 – What type of effect size?

L306 – Reference needed for the effect size interpretation?

Results

L310 – You need to identify the magnitudes of the changes throughout the results, currently you have only reported significance.

Discussion

L353-354 – I would reorder this sentence as it is currently poorly structured.

L354-355- Change to “Both VICAMS and BS showed improvements…”

L356-357 – abbreviation

L358 – reference style

L367-368 – delete this sentence starting with “For this..”

L377 – provide references after previously

L378 – Is decreased muscle-tendon unit stiffness a desired adaptation for athletes who you are trying to apply this research to?

L392 – reference stylte

L394 – Maybe not the main objective, but it would still be a adaptation (even based of previous literature).

L396 – Don’t suggest a relationship as you have not assessed this.

L398 – Is VICAMs inhibition through the golgi tendon organ? Is this why there is a greater increase in AROM for VICAMS? Explain the difference in findings for the BS and VICAMS.

L403 – It is not really strength development after 1 session/acute bout. Is it not just a PAP effect.

L405 – abbreviation.

L410 – Change “acute effect of BS on MVIF”

L413 – Could you add the size of the observed change here, either an effect size, mean difference or % change.

L415 – abbreviate on page 1.

L422 – delete “(MIVC)”

L439-443 – What was the difference in measurement devices? Force plates or jump mats only?

L445 – was this the same measurement device?

L446 – reference needed for PAP

L446 – Could you comment on the fact that increased AROM could be increasing CM depth and TTTO effectively changing jump strategy to improve jump height.

L448 – abbreviation

L450 – further investigation into knee or hip moments would be useful

STEARNS, K. M., R. G. KEIM, and C. M. POWERS. Influence of Relative Hip and Knee Extensor Muscle Strength on Landing Biomechanics. Med. Sci. Sports Exerc., Vol. 45, No. 5, pp. 935–941, 2013.

Umberger, Brian R. MS, CSCS. Mechanics of the Vertical Jump and Two-Joint Muscles: Implications for Training. Strength and Conditioning 20(5):p 70-74, October 1998.

L457-459 – Actually if you look at the magnitude of change only VICAMs could be beneficial as the control group improved to a greater degree than the BS group, this needs further examination.

L469 – why the upper body?

L461 – What about the measurement devices used? Not gold standard. Additionally, the study lacks ecological validity for use with athletes.

L475 – Change “MVIC” for “MVIF” as previously used.

L485 – Check your references, there are a number of errors with missing information such as author names. Moreover, you have cited a youtube video. This is not appropriate in academic writing.

Figure 2-6 – You need to vast improvements on all figures. There are vertical axis titles. Additionally, you should report the individual changes not just the mean differences. Moreover, if you look at the results closely ballistic training had pretty much zero effect, why? Additionally, for the CMJ, the control group improved to a greater magnitude than the ballistic group. Why? Probably measurement error.

See Weissgerber TL, Milic NM, Winham SJ, Garovic VD (2015) Beyond Bar and Line Graphs: Time for a New Data Presentation Paradigm. PLoS Biol 13(4): e1002128. https://doi.org/10.1371/journal.pbio.1002128

Table 1 – You need to report effect sizes for all pre-post differences to understand the magnitude.

---

## Round 0.2 · accepted · Accept

I congratulate the authors on a well improved manuscript. All comments have been successfully implemented. Thank you

Reviewer 1 ·

Basic reporting

All conditions are met

Experimental design

Sounds design and with current updates meets the standard.

Validity of the findings

with current updates it's an improved manuscript.

·

Basic reporting

I am happy with the changes made it is now clearer and reads better throughout.

The literature used was sufficient with a professional structure.

Experimental design

The aim and scope of the article is appropriate for the journal with a relevant research question. With appropriate changes made to the methods.

Validity of the findings

Findings are consistent and valid with good explanation throughout.

Additional comments

I thank the authors for making all changes.